# Skeletal and Dental Morphological Characteristics of the Maxillary in Patients with Impacted Canines Using Cone Beam Computed Tomography: A Retrospective Clinical Study

**DOI:** 10.3390/jpm12010096

**Published:** 2022-01-12

**Authors:** María Elena Montes-Díaz, Alicia Martínez-González, Riánsares Arriazu-Navarro, Alfonso Alvarado-Lorenzo, Nuria Esther Gallardo-López, Ricardo Ortega-Aranegui

**Affiliations:** 1Department of Orthodontics, Faculty of Medicine, CEU Universities, 28668 Madrid, Spain; mariaelena.montesdiaz@ceu.es (M.E.M.-D.); amglez@ceu.es (A.M.-G.); 2Department of Basic Medic Science, Faculty of Medicine, CEU Universities, 28668 Madrid, Spain; arriazun@ceu.es; 3Department of Surgery, Faculty of Medicine, University of Salamanca, 37008 Salamanca, Spain; 4Department of Dental Clinical Specialties, Faculty of Dentistry, Complutense University of Madrid, 28040 Madrid, Spain; negallar@ucm.es; 5Section of Radiology, Faculty of Dentistry, Complutense University of Madrid, 28040 Madrid, Spain; riortega@odon.ucm.es

**Keywords:** palatally impacted canine, CBCT, bone discrepancy, precision dentistry, 3D diagnosis

## Abstract

The aim of this study is to analyze the skeletal and dentoalveolar morphological characteristics of the maxillary in subjects with a unilateral palatally impacted canine using Cone Beam Computed Tomography (CBCT). A retrospective clinical study was conducted of 100 adult patients divided into two groups: one consisting of patients with a unilaterally palatally impacted maxillary canine (GI), with the subgroups in the right and left hemiarches (GI-R and GI-L), and the second, without impacted canine, as the control group (CG). The CBCT measured skeletal variables (maxillary basal width and alveolar crest height) and dentoalveolar variables (inclination of the upper incisor, tooth lengths of incisors and canines, arch length, tooth size and bone dental discrepancy). In skeletal variables, statistically significant differences were found in alveolar crest height (ACH) in all groups and subgroups (*p* < 0.01). In the dentoalveolar variables, there were differences in the angle of the upper incisor (II) and lateral incisor length (LLIL) between the GI and GC and the angle of the upper incisor (II′), arch length (AL′) and arch length-tooth size discrepancy (ATD′) among the GI subgroups (*p* < 0.01). There are skeletal and dentoalveolar differences in patients with unilateral palatally impacted maxillary canines, with lower angular and linear measurements compared with patients without impaction.

## 1. Introduction

Tooth eruption is the vertical movement from the intra-alveolar non-functional position to the functional line of occlusion [1]. The maxillary canine erupts at age 11 in boys and at approximately 10.6 years in girls [2], with a longer development period, deeper area of formation, and a more difficult path to the occlusal position [3,4,5,6]. Coulter and Richardson observed a distance traveled from formation to eruption of 22 mm [7]. It is considered impacted when it is completely or partially covered by mucoperiosteum and bone, distant from the place and time that corresponds to it in the mouth [8,9]. After the third molar, the canine is the tooth that most frequently presents anomalies in its eruption [5,9,10], with a prevalence of 1–3% [2,4,5,6,11,12,13,14,15], and in the majority of the cases, it is an asymptomatic process [14]. Exclusively within orthodontic practice, the prevalence amounts to 23.5% of orthodontic patients [2,16]. The incidence of the location of the impacted canine depends on race [17], with palatal impaction being more frequent [6,9,18,19,20] in 85% [16,21,22], except in the Asian race where buccal impaction is three to six times more common [9,17]. The 83–92% of canine impactions are unilateral [5], with a higher prevalence (58%) on the left side [23]. Numerous studies have affirmed that it clearly predominates in the female sex [2,4,6,7,9,15,17,18,23,24] in a 2:1 ratio [2,6,18].

The first reported cases of canine impaction dates back to prehistory [15]. Its etiology is unclear and appears to be multifactorial, mainly related to loss of lateral incisor guidance and genetics [5,13,17,25,26]. In 2020, Chen et al. suggested that the structural variation of the maxillary bone could constitute an etiological factor of impaction or be the result of the lack of maxillary development [5]. Canine impaction is considered to have a sufficient prevalence to carry out extensive studies of possible preventive treatment modalities [16].

Conventional two-dimensional (2D) radiography has shown errors, and lack of diagnostic precision and unsatisfactory information not only in dental impactions [4,24] but also in the periodontal diagnosis of the alveolar bone [27]. Cone Beam Computed Tomography (CBCT) has become the alternative and the reference diagnostic test to determine the position of impacted canines [2,5,28] because of its reduced distortion, lower cost and lower radiation exposure with respect to other three-dimensional techniques such as computerized tomography (CT) [4] because it allows for obtaining the specific image of the canine area without the need to expose other areas [1,11] Therefore, we can see the exact position, angulation, assessment of adjacent structures and associated root resorptions [2,12]. The kilovoltage of the CT and the CBCT are, respectively, 120 and 60–90 kV [29], therefore obtaining a lower radiation with the second method. Some authors, such as Bjerklin and Ericson, have suggested that the 3D methodology changed almost 30% of the diagnoses of impacted canines originally diagnosed in 2D [30]. The failed and/or late diagnosis of a canine impaction can have important consequences on intraoral health such as root resorption of neighboring teeth [4,10], ankylosis [16,24], aesthetic repercussions, decreased available space in the arch, follicular cyst formation and eventual tooth loss due to periodontal problems [12]. The non-eruption of an impacted canine increases the risk of root resorption of the adjacent tooth, which is usually less than 1 mm apart [12]. Baccetti [31] and other authors [9,12,32] found a relationship between canine impaction and other dental anomalies such as smaller lateral incisors, aplasia of second premolars, infraocclusion of temporary molars and enamel hypoplasia. The literature also reported an association with agenesis in 33% of the cases of impacted canines [4,16], suggesting that these events could have the same genetic origin [12].

The aim of this study was to analyze patients with a unilateral palatally impacted maxillary canine using a 3D method such as CBCT, determine if there were statistically significant differences between the skeletal and dentoalveolar morphological characteristics of the maxilla with respect to a control group and to compare the impacted hemiarch with the non-impacted hemiarch within the impacted canine group. The null hypothesis of the present study is that there are no statistically significant differences between skeletal and dentoalveolar morphological variables in patients with unilateral palatally impacted maxillary canines and patients without impaction.

## 2. Materials and Methods

### 2.1. Ethics Approval and Patient Consent

This work followed the guidelines established by the Declaration of Helsinki for human research, and the project was approved by the ethics committee of the Faculty of Medicine, CEU Universities, Madrid, Spain (ref: 328/19/16). The participants were asked for their written consent.

### 2.2. Sample Size Calculation and Participants

This study was carried out on 100 patients, a sample size similar to other published studies [1,8]. To determine the sample size, previous studies were used as an initial guide. A pilot study of 20 subjects was carried out, and with the difference and deviation observed, the necessary sample size of 47 subjects per group was calculated to obtain a 95% significance difference with a power greater than 80%. The analysis software used for this calculation was SPSS 28 for Windows (Armonk, NY, USA).

### 2.3. Study Design

A retrospective cross-sectional clinical study was carried out on a total sample of 100 Caucasian patients who were divided into two groups: 50 patients with unilateral palatal impacted maxillary canine (GI), where 54% were women (*n* = 29) and 46% men (*n* = 21), and 50 control subjects (CG), where 58% were women (*n* = 27) and 42% were men (*n* = 23). Within the GI group, a split mouth study was performed, dividing the GI into two subgroups: 21 patients with impacted canines on the left side (GI-L group) and 29 on the right side (GI-R group), comparing them with the control side (Table 1).

We compared the GI and GC group values; to compare the subgroups within the GI (GI-L and GI-R), a split-mouth study was performed. The samples were collected over a period of 22 months (June 2019–April 2021) from the same private clinic. The initial sample of this study consisted of 352 CBCT images, of which 100 were selected for the study according to inclusion and exclusion criteria. For the selection of the participants, the following inclusion criteria were followed: patients without growth whose age range was between 20 and 45 years, with unilateral palatal impaction of a maxillary canine, without an interfamilial relationship to other patients in the sample and with adult permanent dentition. The exclusion criteria were bilateral impactions of maxillary permanent canines or canines in the buccal position, severe craniofacial syndromes or malformations, lack of severe sagittal skeletal malocclusions (angle ANB between 0–5°), severe maxillary compression, periodontal disease, previous orthodontic treatment, missing teeth or agenesis (excluding wisdom teeth).

### 2.4. Measurement Procedure

The study variables were measured in a Cone Beam Computed Tomography (CBCT) radiological test, of which the model was the Planmeca Promax 3D MID (Planmeca, Helsinki, Finland), using the Invivo™ 6 KaVo (Kavo, Hatfield, PA, USA) program to perform the measurements. The 3D imaging volume range covered everything from an individual tooth (Ø34 × 42 mm) to obtaining the volume of the entire facial region (Ø160 × 160 mm). The Ø160 × 160 mm volume was used, where a three-dimensional image of the complete skull was obtained. The total exposure time was approximately 18–26 s, with a kilovoltage of 54–90 kV and a milliamperage of 1–14 mA, with a volume reconstruction time of at least 15 s. The posture of the patient during the exposure was with the Frankfurt plane parallel to the ground.

The skeletal variables measured were the following:Basal maxillary width (BMW) in the frontal plane, measured in the JL-JR Ricketts distance in the GC and GI (linear measurement between points on the jugal process at the intersection of the outline of the maxillary tuberosity and the zygomatic buttress) [26] and in the GI-R and GI-L, the JR and JL distance to the median palatal raphe (J-PR) (Figure 1).Maxillary alveolar crest height (ACH) was measured from a tangent to the floor of the nasal fossa, to the lowest alveolar crest between both central incisors (Figure 2) in the GC and GI and, in the GI-R and GI-L, the alveolar crest height from a tangent to the floor of the nasal fossa, to the lowest alveolar crest of the canines, and in its absence, to the lowest alveolar crest of the place that would correspond to the impacted canine (ACH′) [8] (Figure 2).

The dentoalveolar variables were as follows:The angle of the coronal inclination of the upper right and left central incisors with respect to the palatal plane (II) was measured in the sagittal plane (anterior nasal spine-posterior nasal spine) [33] where the angle obtained is that between the major axis of the upper incisor and the palatal plane in the GI and GC. In the GI-R and GI-L subgroups, the coronal inclination of the upper right and left central incisors was measured separately with respect to the palatal plane (II′) (Figure 3).Relative canine length of the upper right (RCL) and left (LCL) canine in its main axis, from the apex to the incisal ridge [34], in the GI and GC. In the GI-R and GI-L, the length of the right and left canine (CL) was measured (Figure 4).Upper right lateral incisor length (RLIL) and left (LLIL) lateral incisor length, along their long axis, from the apex to the incisal ridge [34] in the GI and GC. In the GI-R and GI-L subgroups, the right and left lateral incisor lengths (LILs) were measured (Figure 4).Arch length (AL) from mesial of the first permanent molar on one side to mesial of the first permanent molar on the contralateral side [35] in the GI and GC. In the GI-R and GI-L, from mesial of the first permanent molar on one side to the dental midline and from the dental midline to the mesial side of the contralateral first permanent molar (AL′) was measured (Figure 5).Mesiodistal tooth size (TS) of the entire upper arch except for the first, second and third molars [35] in the GI and GC. In the GI-R and GI-L, the TS of the right and left hemiarches (TS′) was measured (Figure 5).Arch length-tooth size discrepancy (ATD), obtained from the subtraction of the arch length (AL) and mesiodistal tooth size (TS) [35] in the GI and GC. In the GI-R and GI-L subgroups, the ATD of the right and left hemiarches (ATD′) was measured (Figure 5).

### 2.5. Statistical Analysis

The statistical analysis of the data was performed with the SPSS 28.0 program (SPSS Inc., Chicago, IL, USA) for Windows. To determine normal distribution, the Kolmogorov–Smirnov test was used. Once the normal distribution of the data was verified, the paired Student’s *t*-test was used for the comparison of two related samples on the same subject and the Student’s *t*-test of independent samples was used for the comparison between the control and study groups. Two statistical significance levels were established: *p* < 0.05 as statistically significant and *p* < 0.01 as highly significant.

## 3. Results

### 3.1. Intraclass Correlation Coefficient

Measurements were performed by a single operator. The data that reached the statistical operator were kept anonymous by means of a numerical code. To obtain intraoperative agreement, measurements of 10 samples per group were performed, 3 times on 3 different days, with an interval of 3 days between measurements. The time spent measuring each sample was approximately 20 min. From this reassessment, the mean was performed to verify that there was agreement between the measurements. To measure agreement, the Intraclass Correlation Coefficient (ICC) was performed, where 0 was a poor degree of agreement and 1.00 was almost perfect (Table 2).

### 3.2. Differences between the Impacted Group (GI) and Control Group (GC)

In the results obtained for the skeletal variables, significant differences were found (*p* < 0.001) in ACH between the GI and GC (18.52 ± 3.47 vs. 20.80 ± 2.74), with a height 2.28 mm lower than average in the GI with respect to the GC. No significant differences were found in BMW. In the results obtained for the dentoalveolar variables, there were significant differences (*p* < 0.001) in II (101.76 ± 8.31 vs. 106.58 ± 6.61) between the GI and GC, with 4.82° less inclination in the GI with respect to the GC and LLIL (21.14 ± 2.08 vs. 22.33 ± 1.89), with LLIL being 1.19 mm shorter in GI. Significant differences (*p* < 0.05) were found in RCL (25.08 ± 2.54 vs. 26.29 ± 2.37), RLIL (21.24 ± 2.02 vs. 22.14 ± 2.07) and LCL (25.08 ± 2.50 vs. 26.12 ± 2.56), with shorter tooth lengths in the GI compared with the GC in all cases, regardless of the impaction side and in AL (69.37 ± 3.83 vs. 71. 05 ± 3.25), with an average 1.68 mm less space available in the GI with respect to the GC (Table 3). No significant differences were found in TS or ATD.

### 3.3. Differences between the GI-R and Control Side

In the results obtained for the skeletal variables in the GI-R, there were significant differences (*p* < 0.001) in ACH′ (14.87 ± 2.98 vs. 15.40 ± 3.06), 0.53 mm lower than average in the right hemiarch. No significant differences were found between hemiarches in J-PR (Table 4).

In the results obtained for the dentoalveolar variables in the GI-R, statistically significant differences (*p* < 0.001) were found in AL′ (33.72 ± 1.95 vs. 35.09 ± 1.64), with 1.37 mm lower than the mean in the right side with respect to the left side, and in ATD′ (−0.96 ± 1.63 vs. 0.88 ± 2.28), with the mean right side being 1.84 mm more negative than the left. Statistically significant differences (*p* < 0.05) appeared in II′ (99.70 ± 8.08 vs. 101.67 ± 8.48), with the inclination of 1.97° less than the mean in the right hemiarch compared with the left, and in TS′ (34.69 ± 1.71 vs. 34.21 ± 1.90), the tooth size was larger on the right side (with a mean 0.48 mm more than the left TS′). No significance was found in LIL or CL (Table 4).

### 3.4. Differences between the GI-L and the Control Side

In the results obtained for the skeletal variables in the GI-L, significant differences (*p* < 0.001) were found in ACH′ (15.53 ± 3.00 vs. 16.99 ± 3.43), with 1.46 mm lower mean on the left hemiarch. No significant differences were found between hemiarches in J-PR (Table 5).

In the results obtained for the dentoalveolar variables in the GI-L, significance (*p* < 0.001) was found in AL′ (34.14 ± 2.65 vs. 35.80 ± 2.22), with the left side mean 1.66 mm less than the right, and in ATD′ (−1.26 ± 3.08 vs. 0.16 ± 2.40), with the left side −1.42 mm lower. Significant differences (*p* < 0.05) were found in LIL (20.99 ± 2.27 vs. 21.56 ± 1.70), with the left side having a mean of 0.57 mm less. No statistically significant differences were found in II′, CL or TS′ (Table 5).

### 3.5. Differences between the GI-L and GI-R with Control Side

In the results obtained for the skeletal variables in the GI, there were significant differences (*p* < 0.001) in ACH′ (15.15 ± 2.98 vs. 16.07 ± 3.28), where the mean of the impacted hemiarch was 0.92 mm lower. No significant differences were found between hemiarches in J-PR (Table 6).

In the results obtained for the dentoalveolar variables in the GI, there were significant differences (*p* < 0.01) in II′ (100.99 ± 8.42 vs. 102.54 ± 8.69), 1.55° lower in the impacted hemiarch; in AL′ (33.90 ± 2.26 vs. 35.39 ± 1.91), 1.49 mm lower in the hemiarch containing the impacted canine; and in ATD′ (−1.08 ± 2.62 vs. 0, 58 ± 2.00), 1.66 mm more negative in the impacted hemiarch. Significance (*p* < 0.05) was found in LIL (21.01 ± 2.22 vs. 21.37 ± 1.85), with the lateral incisor on the impaction side being 0.36 mm shorter on average. No significant differences were found in CL or TS′ (Table 6).

### 3.6. Differences between Male and Female Sex

Regarding sex, there were significant differences (*p* < 0.01) in one dentoalveolar variable, AL′ (34.88 ± 2.23 vs. 33.19 ± 2.03), with a mean 1.69 mm lower in the female sex. No significance was detected in the rest of the variables (Table 7).

## 4. Discussion

Over the years, factors related to canine impaction have been studied in different diagnostic tests [25,32,36,37], orthopantomographs and plaster casts. In the last decade, CBCT has gained importance for its high level of precision and reliability in linear and angular measurements [1] as well as for being reliable diagnostic method [24,25,37,38].

The present study attempted to analyze and compare the characteristics of the maxilla in patients with a palatal impacted maxillary canine with respect to a control group and with respect to the non-impacted side. In the literature, there are numerous studies that have analyzed bone and teeth characteristics in patients with impacted canines and that have compared the impacted side with the non-impacted side [1,8,12,17,28], but there are fewer studies that have introduced a control group [25,32]. Of all of them, none have compared the impacted side to the non-impacted side within the same study, and the study group compared with the control group, as analyzed in the present study by performing a split-mouth study.

Until a few years ago, it was believed that the permanent upper canine ran out of space because it was the last to erupt in the arch [22]. This usually occurs in both buccal and palatal impactions, with the position of the canine being different in both cases [14,26,36]. In both impactions, a form of crowding is considered the etiology, which is solved with preventive treatment by widening the space that will eventually allow for the eruption of the canine [6,26]; however, palatal impactions cannot always be associated with crowding, which makes the etiology more uncertain. For this reason, we decided to study only palatal impaction.

Most of the published cases of palatally impacted canines are European, with an impaction frequency at least two or three times greater than those buccally impacted [15]. Very few Black cases have been reported, and in Asians, it is quite rare, with a 2:1 ratio of Caucasians to Asians [26]. In Europeans, 70% of the cases of canine impaction are palatal [15,36]. Everything indicates that there are significant differences in the frequency of canine impaction depending on the racial group, which is predominantly Caucasians [5].

Björk and Skieller [39] observed in growing patients that the maxillary posterior basal width increased by 0.4 mm per year between the ages of 4 and 20. Gandini and Buschang [40], in a similar study years later, suggested that the greatest development of the maxilla occurs in late adolescence. For this reason, in this research, the minimum age of the subjects was 20 years, during which growth would have ended or were residual, which was irrelevant to this study, in order to analyze the definitive basal maxillary width. The Ricketts JL-JR distance has been proposed as the anatomical indicator par excellence of transverse dysplasia of the maxilla [26]. Coinciding with Saiar [26], no statistically significant differences were found in the basal maxillary width in any of the groups. There were numerical differences, and it is probable that, by increasing the sample size, statistical significance could appear between the study group and the control group, with narrower jaws in the GI. Si-Chen et al. [5] measured the basal maxillary width with CBCT with the same Ricketts parameter and found narrower maxillae in patients with palatally impacted canines with a smaller sample size than in the present investigation; they probably found significance because their machine learning algorithm used the Learning-based multi-source Integration framework for Segmentation (LINKS). D´Oleo Aracena [8] and Arboleda-Ariza [25] measured the dentoalveolar transverse discrepancy with CBCT in the distance from the groove of the first premolar to the middle palatal raphe in patients with unilateral canine impaction, also finding narrower maxillae in the canine side included. Other authors such as Al-Nimri [20] and Langberg [36] measured the transverse maxillary discrepancy in dental casts in the same way. The first found wider arches on the impaction side, and the second did not find statistically significant differences. Therefore, deficiency in the maxillary transverse arch width could not be a primary factor contributing to the development of palatally impacted canines [36].

Some studies, including that of Wise [41], suggested that alveolar bone growth at the base of the crypt occurs during tooth eruption. In his study on rats, he suggested that the BMP6 gene could be essential in the development of the alveolar bone. CBCT has shown greater morphological details of the alveolar bone with respect to other radiological techniques [27]. In the present study, it was found, coinciding with the study by Tadinada [1] in 2005, that maxillary alveolar crest height at the anterior level was lower in the group with the impacted canine as well as on the impaction side at the level of the alveolar bone of both canines in the study group. His study was the first to measure alveolar dimensions and arch length with CBCT in the same subject comparing both sides, which could justify the non-eruption of the canine and the possible alteration of the BMP6 gene, corroborating the genetic theory as one of the etiological factors in canine impaction.

More retroclined upper central incisors were found in the study group and, within it, in the impaction hemiarch, a typical characteristic of Class II division 2 malocclusions. In 2000, Basdra [42] found a relationship between canine impaction and this sagittal malocclusion. Lüdicke [19], using panoramic radiographs and cephalograms, measured the relationship of the upper incisors SNA angle in canine impaction, finding greater retroinclination in palatal inclusions and results coinciding with those of Basdra [42] and Jacoby [22], observing that Class II division 2 exists in almost half of patients with palatally impacted canines. This suggested that another possible reason for impaction, apart from the anteroposterior problem, was excess space in the palate at the level of the apices of the incisors, combined with a relative coronal crowding at the incisal level, coinciding with our data of more negative bone discrepancy in the impacted hemiarch. The literature reported that the absence of an erupted maxillary canine is related to changes in the labial inclination of the incisors [43], with minor coronal inclinations in the four upper permanent incisors in cases of canine impaction compared with subjects without impaction. In our study, however, the subject’s sagittal occlusal relationship was not considered.

In the present study, we found that the study group had lower tooth lengths of the lateral incisors and canines with respect to the control group, regardless of the impaction side. Coinciding with many authors, including Kim and Karacin [17,44]. we found a decreased length of the lateral incisor on the impaction side. Unlike that on the lateral incisor length, there is little literature on the length of the maxillary canines in cases of canine impaction. Hettiarachchi [45] found that palatally impacted canines had shorter roots and longer crowns than non-impacted ones. Yoojun Kim [17] spoke of small lateral incisors and large canine crowns on the impaction side, and in buccal impactions, on the contrary, larger lateral incisors have been observed [9]. Becker et al. [46] suggested in 1981 that the presence of a lateral incisor with adequate root length, formed at the corresponding time, is an important factor in guiding the canine eruption until its functional occlusion.

In 1983, Jacoby [22] observed 46 occlusal photographs of impacted canines in patients with different characteristics (extraction of premolars, agenesis of lateral incisors, ectopic canine on the contralateral side, etc.), concluding that 85% of the canines impacted by palatine had an adequate arch space available. He discarded any radiographic record claiming distortions, magnifications and limitations inherent to two-dimensionality. In the present study, we obtained contrary results, finding significantly less available space in the arch in the study group compared with the control group, and within the study group on the impaction side. This is probably due to the fact that the measurements were carried out with CBCT in a sample with homogeneous characteristics. In 2005, Tadinada [1] measured arch length with CBCT and obtained the same results as in the present study. Some authors such as Yoojun Kim [9] or Mercuri [32] found no significant differences in the available space between patients with unilateral canine impaction and a control group, perhaps because the measurements were made on plaster casts, and the sample size as well as the age range of the subjects of their study were smaller.

In this study, a significantly more negative bone-dental discrepancy was observed in the impaction hemiarch with respect to the contralateral one. Subjects with the temporal canine present (39 cases) and absent (21 cases) were included indistinctly in the study group, since the non-exfoliation of the temporal canine is the consequence of the dislocation of the canine towards the palate but is not its cause [15]. In 2005, Al-Nimri [20] did not find significant differences in the bone dental discrepancy measured in casts, probably because he could not access the measurement of the mesiodistal diameter of the impacted canine, which we were able to access using CBCT, which according to Kim in 2017 [17], presents a width and volume greater than the erupted contralateral canine.

Most of these studies have not considered the sex of the sample and therefore have not analyzed sexual dysmorphism in canine impaction. In 2017, Capitaneanu [47] observed what other authors had already seen: tooth size in women is significantly smaller than in men, with the canine (especially the lower one) being the tooth with the greatest sexual dysmorphism. Seeman [48] observed that the bones in men were larger than in women, but not necessarily denser, also suggested by Schneider [49] in his analysis of bone density and thickness using CBCT, where he found significant differences in the bone density of the maxillary vestibular cortex, being higher in women. We found differences between men and women, although not significantly in all the variables measured except in the alveolar crest height, with consistently lower values in the female sex, corroborating Seeman’s theory [48]. A significant value was only found in arch length. The increase in the thickness of the female skeleton suggests a more complicated and denser intraosseous path of the canine from its formation to its eruption. In 1949, Dewel [3] observed that impaction of the maxillary canines was more frequent in women than in men, in a ratio of 2:1 [2,6,18]. There are several hypotheses in this regard: difference in growth, development, genetics and priorities; women seeking orthodontic treatment more frequently than men and attending the dentist more frequently, which is why more diagnoses are made in women than in men [8,13].

No study was found that compared the right and left side with their contralateral side without impaction, and very few studies have considered sexual dysmorphism within the sample. Some of the limitations were the sample size [4], the lack of homogeneity of the sample regarding sex and the lack of age limits [8]. Genetic factors, bone density and thickness, or hormonal factors, which have been related to canine impaction, were not considered. In the future, we intend to carry out a more homogeneous study, to equalize the percentage of men and women, to include one more group of growing subjects and to add more variables.

In this study and the therapeutic application of the findings, we found a significantly decreased arch length in all of the groups studied: GI, GI-R, GI-L and impaction side. A distalization of the upper arch would increase this arch length and could facilitate the eruption of the permanent maxillary canine. Retroinclination of upper incisors was found in all groups except the GI-L; a proclination of the upper incisors could allow for the natural eruption of the canine without the need for surgical intervention. The length of the lateral incisor and the alveolar crest height were lower in the group and on the impaction side. The treatment of these two parameters is more complex, but it can serve as a warning in the early detection of palatally impacted canines.

## 5. Conclusions

According to the results obtained, we can conclude that there were statistically significant differences in the skeletal variable of the alveolar crest height, between the GI and GC as well as in the GI subgroups, with consistently lower values in the impaction group and its subgroups. Significantly lower values were found in practically all of the dentoalveolar variables, always lower in the GI compared with the GC and on the impaction side of both subgroups. Sexual dysmorphism in AL′ was lower in the female sex. The results obtained in this study indicate that there are significant differences between the two groups, with factors such as alveolar crest height, smaller tooth lengths and lower angulation of variable upper incisors to be considered in the prediction of palatally impacted maxillary canines.

## Figures and Tables

**Figure 1 jpm-12-00096-f001:**
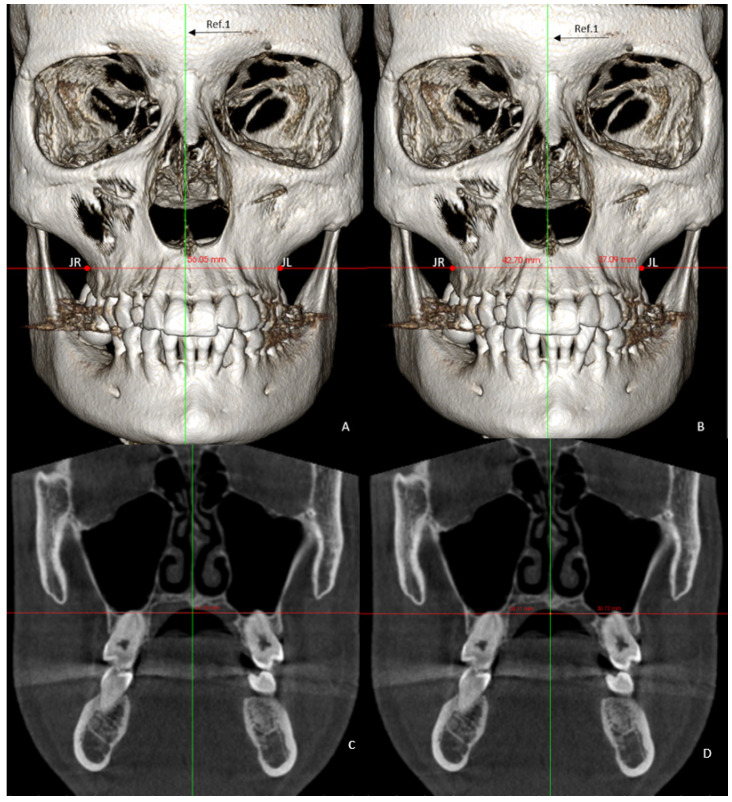
Images of the 3D measurements (**A**,**B**). CBCT reconstruction showing the reference line used for basal maxillary width (**C**) and basal maxillary width per hemiarch (**D**). Ref 1: palatal raphe. JR and JL represent Ricketts’s jugal process.

**Figure 2 jpm-12-00096-f002:**
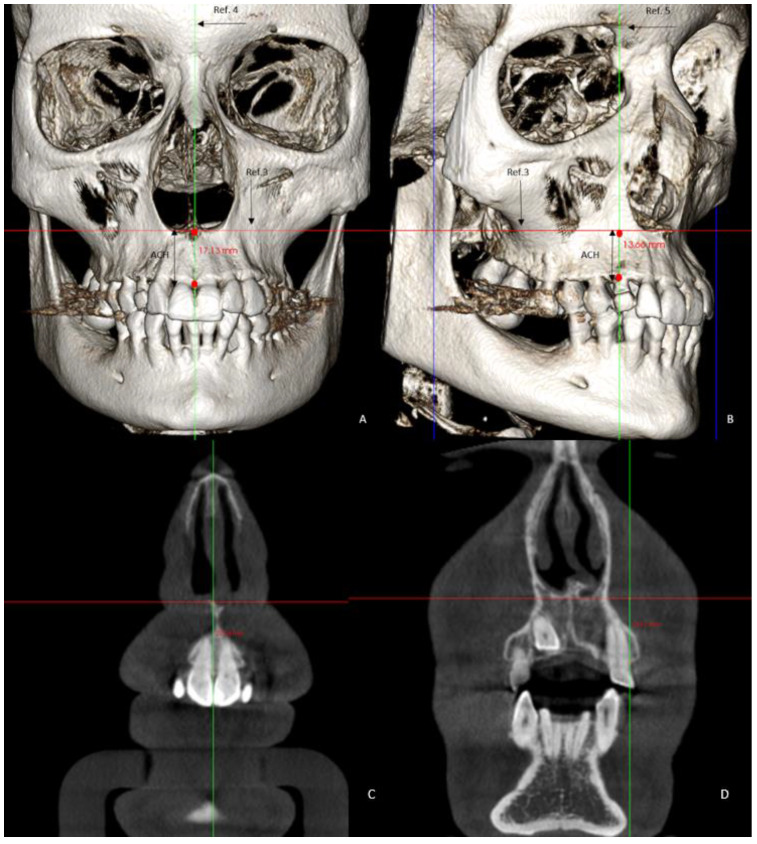
Images of the 3D measurements (**A**,**B**). CBCT reconstruction showing alveolar crest height (ACH) measured from the floor of the nasal fossa to the level of the crest measured between upper central incisors (**C**) and alveolar crest height per hemiarch (ACH′) measured from the floor of the nasal fossa to the level of the crest measured at the level of the right and left canines (**D**).

**Figure 3 jpm-12-00096-f003:**
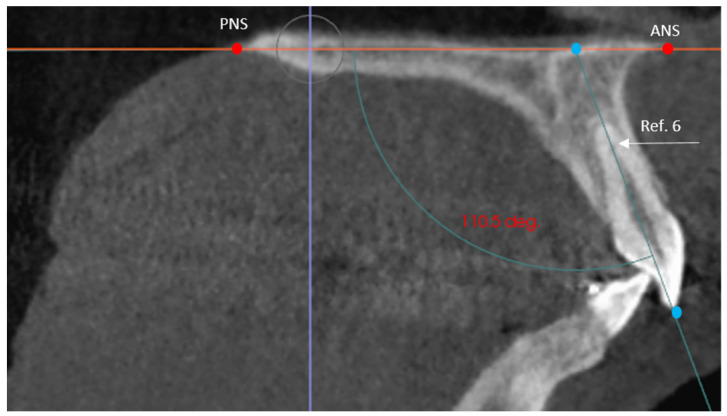
Reference lines used for incisor inclination on the sagittal plane, where PNS and ANS represent the palatal plane and the long axis of the upper incisor (Ref. 6). The angle obtained between both lines defines the incisor inclination.

**Figure 4 jpm-12-00096-f004:**
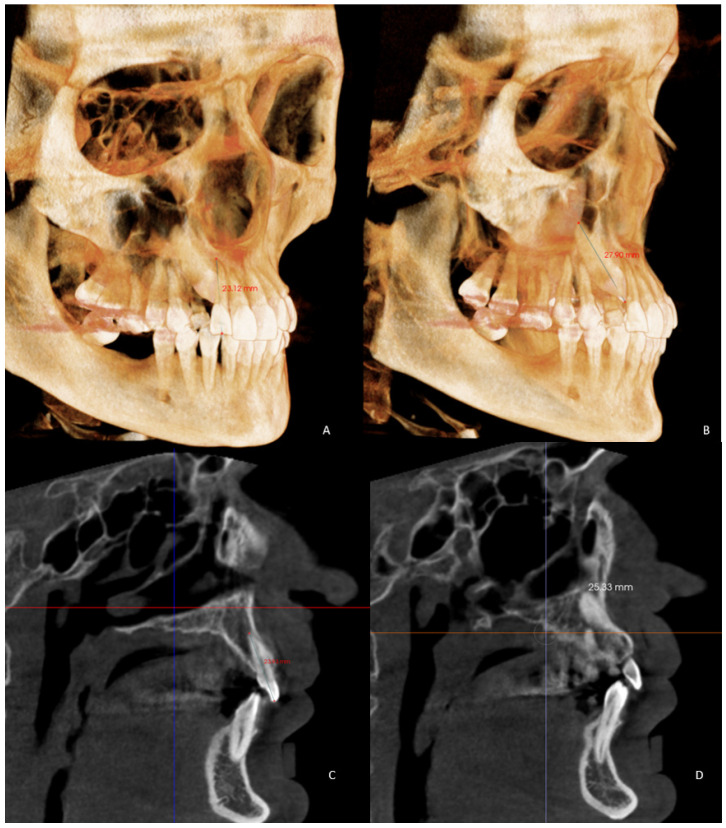
Images of the 3D measurements (**A**,**B**). CBCT reconstruction showing (**C**) total lateral incisor length (LIL) and (**B**) total canine length (**D**).

**Figure 5 jpm-12-00096-f005:**
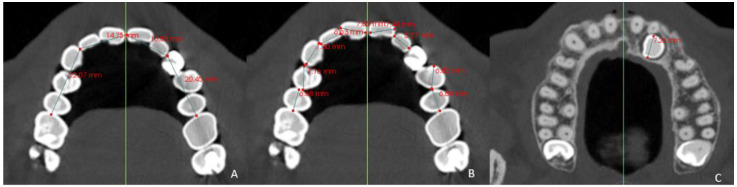
(**A**) Arch length (AL), (**B**) tooth size (TS), and (**C**) tooth size measured in an impacted canine.

**Table 1 jpm-12-00096-t001:** Demographic characteristics of participants (*n* = 100).

			Study Group (GI)	Subgroup Left Side Impaction (GI-L)	Subgroup Right Side Impaction (GI-R)	Control Group (GC)
Total Sample			50	21	29	50

Age	Mean		32.8	30.33	34.55	26.8
	SD		9.16	9.35	9.08	5.78
Sex	Men	*n*	21	12	9	23
		%	46	57.14	31.03	43
	Women	*n*	29	9	20	27
		%	54	42.86	68.96	58

SD: standard deviation; *n* = sample size; % = percentage of simple size.

**Table 2 jpm-12-00096-t002:** Intraclass correlation coefficients for the GC and GI.

Single Measures	Intraclass Correlation
Skeletal Variables	
BMW	0.717
ACH	0.874
Dentoalveolar Variables	
II	0.957
RCL	0.874
RLIL	0.791
LLIL	0.574
LCL	0.870
AL	0.935
TS	0.670
ATD	0.743

GI: Impaction Group; GC: Control Group; BMW: Basal Maxillary Width; ACH: Alveolar Crest Height; II: Incisor Inclination; RCL: Right Canine Length; RLIL: Right Lateral Incisor Length; LLIL: Left Lateral Incisor Length; LCL: Left Canine Length; AL: Arch Length; TS: Tooth Size; ATD: Arch Length-Tooth Size Discrepancy.

**Table 3 jpm-12-00096-t003:** Comparisons of the skeletal and dentoalveolar variables of the GI and GC (*n* = 100).

Variables	GI *n* = 50	GC *n* = 50	*p* Value
Skeletal variables			
BMW (mean ± SD) (mm)	58.39 ± 3.50	59.77 ± 4.14	0.075
ACH (mean ± SD) (mm)	18.52 ± 3.47	20.80 ± 2.74	>0.001 **
Dentoalveolar variables			
II (mean ± SD) (degree)	101.76 ± 8.31	106.58 ± 6.61	0.002 **
RCL (mean ± SD) (mm)	25.08 ± 2.54	26.29 ± 2.37	0.016 *
RLIL (mean ± SD) (mm)	21.24 ± 2.02	22.14 ± 2.07	0.031 *
LLIL (mean ± SD) (mm)	21.14 ± 2.08	22.33 ± 1.89	0.003 **
LCL (mean ± SD) (mm)	25.08 ± 2.50	26.12 ± 256	0.043 *
AL (mean ± SD) (mm)	69.37 ± 3.83	71.05 ± 3.25	0.021 *
TS (mean ± SD) (mm)	69.80 ± 4.24	70.43 ± 3.79	0.433
ATD (mean ± SD) (mm)	−0.42 ± 4.11	0.61 ± 2.90	0.147

* Statistically significant results (*p* < 0.05); ** Statistically significant results (*p* < 0.01). GI: Impaction Group; GC: Control Group; BMW: Basal Maxillary Width; ACH: Alveolar Crest Height; II: Incisor Inclination; RCL: Right Canine Length; RLIL: Right Lateral Incisor Length; LLIL: Left Lateral Incisor Length; LCL: Left Canine Length; AL: Arch Length; TS: Tooth Size; ATD: Arch Length-Tooth Size Discrepancy.

**Table 4 jpm-12-00096-t004:** Comparisons according to the right side (GR) vs. control side (GC´) of impaction of the skeletal and dentoalveolar variables (*n* = 29).

Variables	GI-R *n* = 29	Control Side *n* = 29	*p*-Value
Skeletal variables			
J-PR (mean ± SD) (mm)	40.18 ± 4.08	38.88 ± 3.36	0.126
ACH′ (mean ± SD) (mm)	14.87 ± 2.98	15.40 ± 3.06	0.005 **
Dentoalveolar variables			
II′ (mean ± SD) (degree)	99.70 ± 8.08	101.67 ± 8.48	0.036 *
CL (mean ± SD) (mm)	25.20 ± 2.27	24.97 ± 2.65	0.523
LIL (mean ± SD) (mm)	21.01 ± 2.23	21.24 ± 1.97	0.369
AL′ (mean ± SD) (mm)	33.72 ± 1.95	35.09 ± 1.64	<0.001 **
TS′ (mean ± SD) (mm)	34.69 ± 1.71	34.21 ± 1.90	0.037 *
ATD′ (mean ± SD) (mm)	−0.96 ± 1.63	0.88 ± 2.28	<0.001 **

* Statistically significant results (*p* < 0.05); ** Statistically significant results (*p* < 0.01); J-PR: Jugal Point-Palatal Raphe distance; ACH′: Alveolar Crest Height per hemiarch; II′: Incisor Inclination; CL: Canine Length; LIL: Lateral Incisor Length; AL′: Arch Length per hemiarch; TS′: Tooth Size Length per hemiarch; ATD′: Arch Length-Tooth Size Discrepancy Length per hemiarch.

**Table 5 jpm-12-00096-t005:** Comparisons according to the left side (GI-L) vs. control side (GC´) of impaction of the skeletal and dentoalveolar variables (*n* = 21).

Variables	GI-L *n* = 21	Control Side *n* = 21	*p*-Value
Skeletal variables			
J-PR (mean ± SD) (mm)	40.46 ± 3.28	41.63 ± 4.09	0.211
ACH′ (mean ± SD) (mm)	15.53 ± 3.00	16.99 ± 3.43	>0.001 **
Dentoalveolar variables			
II′ (mean ± SD) (degree)	102.76 ± 8.75	103.73 ± 9.03	0.125
CL (mean ± SD) (mm)	25.24 ± 2.34	24.92 ± 2.93	0.380
LIL (mean ± SD) (mm)	20.99 ± 2.27	21.56 ± 1.70	0.055 *
AL′ (mean ± SD) (mm)	34.14 ± 2.65	35.80 ± 2.22	>0.001 **
TS′ (mean ± SD) (mm)	35.40 ± 2.58	35.63 ± 2.62	0.496
ATD′ (mean ± SD) (mm)	−1.26 ± 3.08	0.16 ± 2.40	0.003 **

* Statistically significant results (*p* < 0.05); ** Statistically significant results (*p* < 0.01). J-PR: Jugal Point-Palatal Raphe distance; ACH′: Alveolar Crest Height per hemiarch; II′: Incisor Inclination; CL: Canine Length; LIL: Lateral Incisor Length; AL′: Arch Length per hemiarch; TS′: Tooth Size Length per hemiarch; ATD′: Arch Length-Tooth Size Discrepancy Length per hemiarch.

**Table 6 jpm-12-00096-t006:** Comparisons of skeletal and dentoalveolar variables of the impacted and non-impacted sides (*n* = 100).

Variables	Imp. *n* = 50	Control Side. Non-Imp. *n* = 50	*p*-Value
Skeletal variables			
J-PR (mean ± SD) (mm)	40.30 ± 3.73	40.03 ± 3.89	0.674
ACH′ (mean ± SD) (mm)	15.15 ± 2.98	16.07 ± 3.28	>0.001 **
Dentoalveolar variables			
II′ (mean ± SD) (degree)	100.99 ± 8.42	102.54 ± 8.69	0.010 **
CL (mean ± SD) (mm)	25.22 ± 2.28	24.95 ± 2.74	0.294
LIL (mean ± SD) (mm)	21.01 ± 2.22	21.37 ± 1.85	0.051 *
AL′ (mean ± SD) (mm)	33.90 ± 2.26	35.39 ± 1.91	>0.001 **
TS′ (mean ± SD) (mm)	34.99 ± 2.12	34.81 ± 2.31	0.347
ATD′ (mean ± SD) (mm)	−1.08 ± 2.62	0.58 ± 2.00	>0.001 **

* Statistically significant results (*p* < 0.05); ** Statistically significant results (*p* < 0.01). J-PR: Jugal Point- Palatal Raphe distance; ACH′: Alveolar Crest Height per hemiarch; II′: Incisor Inclination; CL: Canine Length; LIL: Lateral Incisor Length; AL′: Arch Length per hemiarch; TS′: Tooth Size Length per hemiarch; ATD′: Arch Length-Tooth Size Discrepancy Length per hemiarch. Imp.: Impacted Side; Non Imp.: Non-impacted Side.

**Table 7 jpm-12-00096-t007:** Comparisons according to sex of the skeletal and dentoalveolar variables of the GI.

Variables	Male *n* = 21	Female *n* = 29	*p*-Value
Skeletal variables			
J-PR (mean ± SD) (mm)	41.37 ± 3.29	39.52 ± 3.89	0.084
ACH′ (mean ± SD) (mm)	19.10 ± 3.73	19.39 ± 3.23	0.769
Dentoalveolar variables			
II′ (mean ± SD) (degree)	101.98 ± 8.65	100.27 ± 8.34	0.485
CL (mean ± SD) (mm)	25.56 ± 2.35	24.97 ± 2.22	0.368
LIL (mean ± SD) (mm)	21.58 ± 1.52	20.49 ± 2.56	0.124
AL′ (mean ± SD) (mm)	34.88 ± 2.23	33.19 ± 2.03	0.007 **
TS′ (mean ± SD) (mm)	35.59 ± 1.88	34.55 ± 2.21	0.089
ATD′ (mean ± SD) (mm)	−0.70 ± 2.41	−1.36 ± 2.77	0.387

** Statistically significant results (*p* < 0.01). GI: Impacted group; J-PR: Jugal Point-Palatal Raphe distance; ACH′: Alveolar Crest Height per hemiarch; II′: Incisor Inclination; LIL: Lateral Incisor Length; CL: Canine Length AL′: Arch Length; TS′: Tooth Size; ATD′: Arch length-Tooth Size Discrepancy.

## Data Availability

The data are available upon request due to privacy and ethical restrictions.

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
