# Peer review of "Skeletal and Dental Morphological Characteristics of the Maxillary in Patients with Impacted Canines Using Cone Beam Computed Tomography: A Retrospective Clinical Study"

_jpm, 2022, doi:10.3390/jpm12010096_

Round 1

Reviewer 1 Report

commented are mentioned in the below file 

Reviewer 2 Report

I congratulate for your manuscript and scientific work!

My remarks: 

  1. In Materials and methods please review the proper name of the CBCT device:  Planmeca Promax roMX 3D MID
  2. In Materials and methods please review the proper company info of Invivo™ 6 KaVo (KaVo, Berlin, Germany)
  3. In Materials and methods "nostrils" refer to a soft tissue anatomical landmark. The measurements were carried out on hard tissue anatomical landmarks. Please clarify!
  4. The skeletal variables were measured on 3D rendered images, not on the reconstructed CBCT images according to the Figure 1, 2 and 4. If it is true, the validity of measurements on 3D rendered images needs to be stated in the manuscript and supported by other studies from the scientific literature. If the measurements were carried out on the reconstructed CBCT images, then it needs to be indicated, that the Figures of 3D rendered images are just demonstrating the mesurement for a better understanding.
  5. Table 2 showing results in Materials and Methods section. Please revise it's proper position in the manuscript! 
  6. The several spelling or spacong error shall be revised and corrected.